# Failure Mode Detection and Validation of a Shaft-Bearing System with Common Sensors

**DOI:** 10.3390/s22166167

**Published:** 2022-08-17

**Authors:** Chung-Hsien Kuo, Yu-Fen Chuang, Shu-Hao Liang

**Affiliations:** 1Department of Mechanical Engineering, National Taiwan University, Taipei 106216, Taiwan; 2Department of Electrical Engineering, National Taiwan University of Science and Technology, Taipei 106335, Taiwan; 3Industry 4.0 Implementation Center, National Taiwan University of Science and Technology, Taipei 106335, Taiwan

**Keywords:** failure mode detection, rolling bearing, PCA, WPD, LSTM

## Abstract

Failure mode detection is essential for bearing life prediction to protect the shafts on the machinery. This work demonstrates the rolling bearing vibration measurement, signals converting and analysis, feature extraction, and machine learning with neural networks to achieve failure mode detection for a shaft bearing. Two self-designed bearing test platforms with two types of sensors conduct the bearing vibration collection in normal and abnormal states. The time-domain signals convert to the frequency domain for analysis to observe the dominant frequency between these two types of sensors. In feature extraction, principal components analysis (PCA) combines with wavelet packet decomposition (WPD) to form the two feature extraction methods: PCA-WPD and WPD-PCA for optimization. The features extracted by these two methods serve as input to the long short-term memory (LSTM) networks for classification and training to distinguish bearing states in normal, misaligned, unbalanced, and impact loads. The evaluation arguments include sensor types, vibration directions, failure modes, feature extraction methods, and neural networks. In conclusion, the developed methods with the typical lower-cost sensor can achieve 97% accuracy in bearing failure mode detection.

## 1. Introduction

Rolling bearings are the standard and broadly used parts on rotating machinery. They are usually subjected to the load in the axial and radial direction with variable load changes during operation. Misalignment, unbalanced, and impact loads are the commonly seen abnormalities on the machines, which can cause bearing life to decrease and can affect the relevant components, resulting in severe damage, especially under high speed, heavy loading, and long duration. Moreover, the bearing’s torque and radial internal clearance affect the machinery’s performance and life. Tong V.-C et al. (2018) [1] have studied the torque of angular contact ball bearings subjected to angular misalignment, and Am-brożkiewicz et al. (2022) [2] present the effects of radial internal clearance related to misalignment. Both demonstrate that the misalignment containing diverse factors can make the failure even more severe.

Bearing vibration detection on the rolling elements of the machines could be the essential method for machine diagnosis and prognosis (Ian et al., 1994) [3]. Prediction of the bearing failure provides early warning for maintenance, can prevent unexpected interruption in operation and can prolong the lifetimes of the machines. The rolling element bearing feature extraction and anomaly detection based on vibration monitoring offer a paradigm for sensor settlement and feature extraction (B. Zhang. 2008) [4]. Both articles mentioned above investigated the effects of bearing vibration for abnormality detection to improve the smoothness of machine operation by evaluating bearing health and minor fault characteristics. Unbalance and misalignment in rotor bearings are essential for studying machine vibration effects. Mogal and Lalwani [5] introduced an order analysis method to investigate the significant impacts of these abnormalities on machines.

Jin et al. [6] define the three life stages of a bearing as RUL: run-in, useful life, and wear-out. They conducted the experiments via abnormality classification to offer an early warning before leading to worse conditions. A nonlinear model was built to track the bearings’ degradation process.

The accelerating aging process for bearing lifetime prediction or long-term practical failure experiments can be costly, time-consuming, and unable to predict future application scenarios. In recent relevant developments, artificial intelligence technology seems more superficial, efficient, and economical and can be even more accurate. Therefore, more and more studies have applied machine learning for fault prediction.

An automatic bearing fault diagnosis project adopted the one-class support vector machines (SVMs), which can automatically find a decision boundary to determine whether a new data point is similar to the training data. The isolation trees and one-class SVMs are used in the machine learning algorithm to identify abnormal points [7].

In terms of extracting signal features, the study of the classification of the alcoholic electroencephalograph (EEG) uses wavelet packet decomposition (WPD) and principal component analysis (PCA) to build the neural network model by M. Saddam et al. [8]. The result reveals that the PCA helps to reduce the data dimension in the computing process and improves accuracy over the other methods. We think combining WPD and PCA can be feasible and even more effective than using only one feature extraction method.

The practice of deep learning on bearing inspection includes data feature extraction, selection, and classification. Many studies used deep learning to achieve better results, particularly in establishing time series models and solving forecasting tasks. For example, B. Li et al. applied a neural network to motor rolling bearing fault diagnosis, demonstrating specific classification processing capabilities for nonlinear problems. The training results show that the model can effectively distinguish the different causes of bearing vibration [9]. Different neural networks are also often used as research objects to compare which neural network is more efficient or more accurate. For example, a study by Al-Raheem et al. of the rolling bearing diagnostics by three artificial neural networks (ANNs), including RBF NN, MLP-BP, and PNN, performs the classification problems compared with the Laplace wavelet analysis method. The MLP-BP can achieve reasonable classification success rates, but its training time is longer than that of the PNN [10].

Similarly, Levent Eren proposed one-dimensional convolutional neural networks for bearing fault detection to compare with the popular MLP, RBFN, and SVM classifier algorithms. The CNN model has better accuracy than the others in the proposed fault detection models [11]. The time spent on computing was a significant index in selecting neural network algorithms.

Zhang et al. proposed a fault diagnosis model based on a deep neural network (DNN). That model omitted signal processing and fault feature selection and used the original time series signal data directly as inputs to train the deep neural network model [12]. Based on the abovementioned articles, the sophisticated neural network can perform better prediction accuracy but simultaneously takes more computing resources and time. Thus, the algorithms that can reduce the computing resource requirements have started to apply the prediction models, such as long short-term memory (LSTM). The LSTM network is a type of RNN (recurrent neural network) that uses the addition of special units to the standard units. The LSTM units include a “memory cell” that can maintain information in the memory for long periods. A set of gates controls the information that enters the memory when it is outputted and forgotten [13].

Long-term sequence management algorithms are suitable for analyzing long-term sequence data sets, such as continuous bearing vibration data acquisition. Pan et al. performed a model that combined CNN and LSTM recurrent neural networks to diagnose bearing faults [14]. That experiment utilized a CNN for automatic feature extraction from high-dimensional data with fair accuracy and was then conducted with the LSTM to consider time coherence for accurate classification. Ultimately, the LSTM method achieved good fault classification with up to 99% accuracy.

Gers et al. applied LSTM to time series prediction through a time-window approach. They found that the LSTM’s ability to track slow oscillations in the chaotic signal may apply to cognitive domains such as rhythm detection in speech and music [15]. In addition, Xu et al. used an LSTM algorithm with long data sequences as the time dimension feature of the extraction of the time series data and combined the LSTM algorithm with a generative adversarial network (GAN) to extract deep features from average bearing vibration data [16]. That model achieved good feature extraction and abnormal state classification with the time series data, demonstrating the power and adaptability of long-term series algorithms.

Summarizing the algorithms above, a suitable feature extraction method that combines a neural network with an effectiveness-processing dataset would achieve outstanding performance on prediction abnormality modes. Thus, we adopted wavelet packet decomposition (WPD), principal component analysis (PCA), and the LSTM algorithm for feature extraction and the neural network model.

Failure mode detection is essential for bearing life prediction to protect the shafts on the machinery. We explore relative technologies to improve failure detection accuracy, which can be applied to bearing or machine lifetime prediction. This work demonstrates the rolling bearing vibration measurement, signals converting and analysis, feature extraction, and machine learning with neural networks to achieve failure mode detection for a shaft bearing.

With the advent of the internet of things, deploying large-scale sensors on machines may become the norm in most manufacturing fields. Additionally, the trend toward intelligent approaches has led to technologies intending to use simple components with intelligent algorithms to provide cost-effective solutions. That inspired us to conduct this study. This work explores failure mode detection in shaft-bearing systems with common sensors, which can be acquired easily at a lower cost.

The introduction presents the relevant research, principles, and ideas that inspire us. The principles and theory section addresses the mathematical principles, algorithms, and neural networks applied to vibration signals processing. The experiments and methods section depicts the apparatus used for data collection, including the specifications and parameters. The results and discussion section shows the graphics, charts, and data regarding the vibration signals, frequency spectra, feature extraction segments, and model training results. Lastly, the conclusion sums up the accomplishments of this work.

## 2. Principles and Theory

The vibration signals from the sensors on the bearing reflect the physical loading changes over time. Converting time-domain data into frequency-domain data can classify the specific characteristics in dominant and harmonic frequencies. Furthermore, we plan the data processing by referring to the studies mentioned above in the introduction. The feature extraction process combines the advantages of PCA and WPD to develop new methods. In the training mode, LSTM seems more committed to the other ANNs, based on reviewing the relevant works in the last section. In addition, the dataset samples for bearing vibration give a typical paradigm for data collection.

### 2.1. Bearing Characteristic Frequencies

Bearing state detection is primarily based on the bearing characteristic frequencies, and the amplitude of the peaks determines whether there is damage to a specific element or not. In any case, the characteristic frequencies are observable in the frequency spectra. The information in the vibration signals can be used to analyze the causes of bearing failures. The extraction of bearing signals involves collecting and extracting features from the signals and making decisions based on these features. 

The outer raceway of the ball-type bearing is assumed to be fixed, and only the inner raceway rotates with the shaft. The associated calculation based on the structure and size of the bearing is shown in the following equations, which include BPFI, BPFO, FTF, and BSF (1)–(4) [17]: 

BPFI: (ball pass frequency, inner race)
(1)$$\;\;\;\;\;\;f_{i}=\frac{N}{2}\left(1+\frac{d}{D}\cos\alpha \right)f_{r}=\frac{8}{2}\left(1+\frac{5.6}{20}\cos0\right)\frac{2800}{60}=238.9\left(Hz\right)$$

BPFO: (ball pass frequency, outer race)
(2)$$\;\;f_{o}=\frac{N}{2}\left(1-\frac{d}{D}\cos\alpha \right)f_{r}=\frac{8}{2}\left(1-\frac{5.6}{20}\cos0\right)\frac{2800}{60}=134.4\left(Hz\right)$$

FTF: (fundamental train frequency, the cage speed)
(3)$$\;\;\;\;\;\;f_{d}=\frac{1}{2}\left(1-\frac{d}{D}\cos\alpha \right)f_{r}=\frac{1}{2}\left(1-\frac{5.6}{20}\cos0\right)\frac{2800}{60}=17.03\left(Hz\right)$$

BSF: (ball (roller) spin frequency)
(4)$$f_{b}=\frac{D}{2d}\left(1-{\left(\frac{d}{D}\cos\alpha \right)}^{2}\right)f_{r}=\frac{20}{11.2}\left(1-{\left(\frac{5.6}{20}\right)}^{2}\cos^{2}0\right)\frac{2800}{60}=76.8\left(Hz\right)$$
where N is the number of rolling elements, d is the rolling element diameter, D is the bearing pitch diameter, α is the contact angle, and $f_{r}$ is the bearing rotation frequency in rpm. 

The abnormal characteristic frequency of the bearing is composed of a series of pulses, which are generated at the position of the abnormal parts whenever the bearing rotates. Therefore, many signal characteristic extraction techniques have been developed for rolling bearing fault detection.

### 2.2. Fast Fourier Transform (FFT)

The Fourier transform is one of the most widely used methods in traditional frequency analysis. When a signal is in the time domain, it can be converted into a frequency-domain signal using a Fourier transform. Because signal characteristics can be more prominent in the frequency domain than in the time domain in terms of their characteristics and observability, frequency-domain analysis has gradually developed into a more commonly used signal analytical method. Most signals processed by signal processing are discrete signals rather than continuous domains. A fast Fourier transform (FFT) is an algorithm that computes a sequence’s discrete Fourier transform (DFT) [18].

### 2.3. Feature Extraction

Commonly used feature extraction techniques for signal processing are based on statistical analyses, such as wave crest, root mean square value, and mean value. Thus, statistical analyses cannot effectively minimize the noise interference caused by other factors (e.g., environment, gears). Thus, other feature extraction techniques have been developed to overcome these issues. For example, fast Fourier transform from the time domain to the frequency domain and wavelet transforms is used to extract feature signals, and many methods have tried to use deep learning for training and extracting bearing feature signals [12]. Furthermore, there are rich sources of knowledge about the diagnostics features used in the diagnostics of rotating machines, according to Sharma V. et al. (2016). It presents the condition indicator (CI)-based diagnosis technique, which summarizes various condition indicators for fault diagnosis [19].

Wavelet analysis has proved its excellent capabilities in decomposing, denoising, and signal analysis. It can analyze non-stationary signals and detect transient feature components, which other methods could not perform since wavelets can concurrently impart time and frequency structures [20]. 

Wavelet transform (WT) gives good time and poor frequency resolution at high frequencies and good frequency and poor time resolution at low frequencies. Analysis with wavelets involves breaking up a signal into shifted and scaled versions of the original (or mother) wavelet, i.e., one high-frequency term from each level and one low-frequency residual from the last level of decomposition.

In numerical analysis and functional analysis, there are several popular wavelet transforms, such as continuous wavelet transform (CWT), discrete wavelet transforms (DWT), wavelet packet transform (WPT), and wavelet packet decomposition (WPD). WPD is more effective than WPT because it can decompose not only the low-frequency part but also the high-frequency. 

#### 2.3.1. Wavelet Transform

Wavelet transformation refers to using a finite-length or fast-decaying mother wavelet to represent the signal, which is scaled and translated to match the input signal. The mother wavelet, which is also known as a basic wavelet, can be defined as [21]:(5)$$\psi_{a,b}\left(t\right)=\frac{1}{\sqrt{a}}\psi \left(\frac{t-b}{a}\right)$$
where a is a scaling factor. When $\left|a\right|<1$, the mother wavelet is compressed and has a small degree of support on the time axis, which corresponds to the high frequency because the mother wavelet becomes narrower and changes faster. When $\left|a\right|>1$, the mother wavelet becomes wider and changes more slowly, corresponding to the low frequency. Moreover, in (10), b is the translation parameter used to determine the position of the mother wavelet. The contrast between the wavelet transform and the Fourier transform is that the infinite trigonometric function base is replaced with a finite-length attenuated wavelet base. The wavelet transform formula can be written as:(6)$$X\left(a,b\right)=\frac{1}{\sqrt{b}}\int_{-\infty }^{\infty }x\left(t\right)\Psi \left(\frac{t-b}{a}\right)dt$$

Assuming that there is a time-domain signal $x\left(t\right)$, a is a scale parameter, and b is a translation parameter; thus, the wavelet transform can project to the time scale that takes advantage of the Fourier transform. The frequency of both signals in the frequency domain and their position in the time domain can be known, indicating that the time-frequency spectrum analysis can be performed. 

Wavelet transformation aims to decompose the original signal as the input signal into high- and low-frequency components via orthogonal wavelet decomposition and uses the obtained low-frequency part as the input signal to perform another wavelet decomposition to obtain the next high- and low-frequency components. However, when analyzing time-frequency spectrum localization, the wavelet transform only decomposes low-frequency signals in the decomposition process but does not decompose high-frequency signals; thus, its frequency resolution decreases as the frequency increases. Therefore, the wavelet packet decomposition analytical method should be used.

Wavelet packet decomposition (WPD) can usually be called wavelet packet, sub-band tree, or optimal sub-band tree structuring. As a more sophisticated analytical method of signal decomposition, the wavelet subspace is further decomposed in a binary manner, and the resulting time-frequency planarization is more detailed; thus, the resolution of the high frequency of the signal is also improved. Assuming that *φ*(*t*) is a scaling function and *ψ*(*t*) is a wavelet mother function, where *μ*_0_(*t*) = *φ*(*t*) and *μ*_1_(*t*) = *ψ*(*t*), the relevant formula can be written as follows [22]:(7)$$\mu_{0}\left(t\right)=\sqrt{2}\underset{k}{\sum }h_{k}\mu_{0}\left(2t-k\right)$$
(8)$$\mu_{1}\left(t\right)=\sqrt{2}\underset{k}{\sum }g_{k}\mu_{0}\left(2t-k\right)$$

Then:(9)$$\mu_{2n}\left(t\right)=\underset{k}{\sum }h_{k}\varphi_{n}\left(2t-k\right)$$
(10)$$\mu_{2n+1}\left(t\right)=\underset{k}{\sum }g_{k}\varphi_{n}\left(2t-k\right)$$
where *t* is the time, *k* is the time translation parameter, $g_{k}$ is the low-pass filter parameter, $h_{k}$ is the high-pass filter parameter, and *μ*(*t*) is the wavelet packet.

As the explanation of the wavelet packet decomposition above, based on the wavelet transform of each signal decomposition and the decomposing of the low-frequency component, this method also decomposes the high-frequency component. Therefore, compared with the wavelet transform, which only decomposes the low-frequency components, wavelet packet decomposition can achieve a more acceptable resolution in a complete signal, including the high and low frequencies. 

Figure 1 shows the process of a three-layer wavelet packet decomposition hierarchically. The S at the top represents the original signal, which splits into a1 and d1 blocks as the first layer, where a1 represents the low-pass filter signal, and d1 represents the high-pass filter signal. Then, a1 decomposes into aa2 and ad2 to form the second layer in the same manner as the creation of the first layer. Lastly, it escalates to decompose all the segments in the second layer to form the eight segments in the third layer. These eight segments, S1 to S8 in sequence, are the features used in the training process.

#### 2.3.2. Principal Component Analysis (PCA)

Principal component analysis (PCA) is a linear dimensionality reduction method widely used in machine learning and statistics to analyze data, reduce data dimensionality, and disassociate. Dimensionality reduction is a type of unsupervised learning; as the name suggests, its purpose is to reduce multidimensional series. Most features can be preserved without any information loss in the process. Thus, presenting data relatively concisely is one of the goals of PCA.

PCA is a well-known statistical technique widely applied to solve critical signal-processing problems, such as feature extraction, signal estimation, detection, and speech separation. From the perspective of machine learning, the purpose of PCA dimensionality reduction is to make the engine of classification performance more effective, reduce data complexity, and, most importantly, shorten training time. From a mathematical perspective, the first step of PCA dimensionality reduction must average the original data and take the target covariance matrix [8].

### 2.4. Long Short-Term Memory (LSTM)

Long short-term memory (LSTM), first proposed in 1997, is a neural network model derived from a recurrent neural network (RNN) [13]. LSTM solved the problems of RNN disappearing in a gradient [23] and achieved higher processing efficiencies than RNN [18]. Thus, it is often used to solve problems related to predicting time series.

Figure 2 illustrates the most significant difference between LSTM and RNN. The typical RNN architecture has only one layer, including an activation function. In an LSTM layer, the neurons have three more control gates (input, forget, and output) and the memory cell (the most critical part). The function of the memory cell is to remember the results of the previous time series, and the forget gate controls whether to retain the contents of the previous memory space.

The basic architecture of an LSTM includes four inputs, one output, and one memory storage unit, and each input has its weight. The LSTM neural unit receives the current input ($x_{t}$) and the output from the previous moment ($h_{t-1}$) with two vectors, where W is weight and b is bios. 

For example, Formulas (11)–(14) describe the previous output matrix ($h_{t-1}$) and the current input matrix ($x_{t}$) multiplied by its own weight W and determined by the activation function. The result calculated by Formulas (11), (12), and (15) is primarily determined by the activation function whether to control the value after the calculation to be recorded or not, such as formula (11), which represents whether the value stored in the memory cell can be cleared or stored.

Formula (14) represents the content to be memorized ($C_{t}$) in the current latest memory storage unit, multiplying the previously memorized content ($C_{t-1}$) by the newly calculated ($f_{t}$) and adding the new input matrix ($z_{t}$), which multiplied by the newly calculated ($i_{t}$) and obtains the result ($C_{t}$). Finally, the new memory unit ($C_{t}$) is multiplied by an activation function, and the calculated ($o_{t}$) for the output judgment, to obtain the final output result ($h_{t}$), which can be referred to in Formulas (11)–(16):(11)$$f_{t}=\sigma \left(W_{f}\cdot \left[h_{t-1},x_{t}\right]+b_{f}\right)\;$$
(12)$$i_{t}=\sigma \left(W_{i}\cdot \left[h_{t-1},x_{t}\right]+b_{i}\right)\;$$
(13)$$z_{t}=tanh\left(W_{z}\cdot \left[h_{t-1},x_{t}\right]+b_{z}\right)\;$$
(14)$$C_{t}=f_{t}\cdot C_{t-1}+i_{t}\cdot z_{t}\;\;\;\;\;$$
(15)$$o_{t}=\sigma \left(W_{o}\left[h_{t-1},x_{t}\right]+b_{o}\right)\;\;\;$$
(16)$$\;h_{t}=o_{t}\cdot tanh\left(C_{t}\right)\;\;\;\;\;\;\;\;\;\;$$

### 2.5. Dataset 

The neural network algorithms for bearing anomaly detection require datasets for training and validation. These datasets are collected from a real machine or a testing platform with precise instruments. Many researchers provide excellent examples of the selection of the accelerator and DAQ, sensor types, number of sensors, fault type mode, and characteristics for reference, as Table 1 shows. Some of these databases are available for artificial intelligence researchers to download and perform the exercises. These vibration signal datasets can offer essential materials for researchers in machine learning and artificial intelligence, with machine fault diagnosis saving time on data collection. 

The 6th dataset item in Table 1 is generated by the self-design platform, as described in Section 3, the experiments and methods section. We tried to emulate four operation modes: normal, misalignment, unbalanced, and impact loads, and assumed that signal noise could be near enough to zero to ignore.

Table 2 presents the total dataset collected from the sensors ADcmXL3021 and Hi229. The raw data samples of the normal state for each sensor are about 65 kilo and 50-kilo points. The total number of vibration data collected in the other abnormal states is in the rest of Table 2. All the data were divided into three groups, training (50%), testing (30%), and validation (20%), for data analysis, such as classification, feature extraction, and model training.

## 3. Experiments and Methods

The experiments and methods present three main subjects: the self-design platform, apparatus and sensor settlement, and data processing flow. Two self-design platforms are built for the vibration signals collection, one for the normal state and the other for abnormal conditions. The apparatus parameters and sensor locations illustrate the central architecture of the data collection system. The last, data processing, describes the methods developed for finding the best results. 

### 3.1. Self-Design Platform

The purpose of the self-design platform is to collect the vibration signals of a rolling bearing under the states of normal and abnormal (misalignment, unbalanced, and impact). Figure 3a,b show the platform dimension and bearing location and the illustration of different loads, respectively. Table 3 shows the nominal bearing dimension of the bearing used in both platforms, and the loadings applied on both platforms are listed in Table 4.

Figure 4a,b show the platforms for collecting the vibration data. The first platform (Platform 1) measures the vibration signal of the bearing under the normal state. The second platform (Platform 2) is slightly sophisticated compared to Platform 1, but the main structure is the same as Platform 1. Figure 4a, Platform 1, illustrates a shaft that links the motor, wheel, and bearing support. 

Figure 4b, Platform 2, has a slider that can move freely on the track vector to create a 1.29° misalignment on the shaft. A 51.54 g aluminum cylinder mounted on the wheel creates an unbalanced load regarding the force generated when the shaft rotates. The impact force relied on a motor-driven external object periodically impacting the base, not illustrated here. The power source is a stepper motor, which can run a maximum speed of 2800 rpm, model BLM4203B, manufactured by NiMotion. The motor runs at 300 rpm in experiments regarding the abovementioned theory of bearing anomaly feature frequency in Section 2. 

The impact load test is to understand the difference in the force influence on the vibration signal between the external force and the force generated by the weight of the shaft and wheel. Many studies have not considered the external impact force in bearing life prediction because the external impact can be irregular and unpredictable. Moreover, the external impact force suffered by the machine can be a combination of multiple impact sources in a practical circumstance. The continuous impact demonstrated here is much simpler than real. 

### 3.2. Apparatus and Sensor Settlement 

The data acquisition apparatus consists of a signal board computer (SBC), a Raspberry Pi B4, and two sensors, ADcmXL3021 and Hi229, as shown in Figure 5. Sensor 1, ADcmXL3021, is a type of high-performance MEMS vibration sensor produced by Analog Devices, and it communicates with SBC via the serial peripheral interface (SPI). Sensor 2, Hi229, is a conventional 9-axis IMU sensor with a lower cost produced by HPNUC, and it transmits data to the SBC via a serial port. Both communication ports can provide sufficient speed for data collection. 

The ADcmXL3021 sensor has four signal collection modes: RTS, MTC, MFFT, and AFFT; the MTC mode was adopted in the test and can collect 4096 data points under 3439 Hz over 1.19 s. On the other hand, the Hi229 sensor can collect 400 data points at a rate of 400 Hz over 1 s. The specifications of the ADcmXL3021 and Hi229 are referred to in Table 5. 

Both sensors contain embedded three-axis accelerometers (x, y, z axes) for detecting vibration. The AdcmXL3021 (Sensor 1) and the Hi229 (Sensor 2) are attached to the top and the front side of the bearing support, respectively, as shown in Figure 6. The center of the sensors aligns with the center line of the shaft. The ADcmXL3021 z-axis and the Hi229 x-axis engage the horizontal direction. Meanwhile, the ADcmXL3021 y-axis and the Hi229 y-axis respond to the vibration in the vertical direction. 

### 3.3. Data Processing

Explained here are the data analysis flow, methods, and algorithm development. The data processing covers vibration data collection (normal and abnormal), Fourier transformation, feature extraction, dataset split (training and test), model training, and validation, as illustrated in Figure 7. The theory and principle of the data acquisition algorithm and apparatus in the experiments refer to those mentioned above in the last sub-section. The LSTM model build arguments are provided in Section 3.3.2. 

#### 3.3.1. Feature Extraction

The feature extraction process combines two algorithms: wavelet packet decomposition (WPD) and principal component analysis (PCA); two methods were proposed, method 1: PCA-WPD and method 2: WPD-PCA. Method 1 starts with PCA to reduce the data dimension and perform the three-layer WPD. Conversely, method 2 performs the three-layer WPD first, then reduces the data dimension with PCA. Figure 8a,b illustrate the feature extraction flow of method 1: PCA-WPD and method 2: WPD-PCA, respectively.

Method 1: PCA-WPDThe process starts from PCA (dimension reduction); the bearing raw data for every ten samples can obtain one new sample. Then, using three-layer wavelet packet deposition, it generates eight sub-band segments and calculates these segments separately to create new features. Finally, the process comes to feature normalization after processing the reconstructed decomposition coefficients and building the wavelet packet energy spectrum feature to complete the feature extraction, as shown in Figure 8a. Method 2: WPD-PCASimilarly, Figure 8b illustrates the data processing of WPD-PCA that starts with the three-layer wavelet packet decomposition. Reconstructed decomposition coefficients and built wavelet packet energy spectrum features are conducted after the eight sub-band segments are generated and calculated. Performing dimension reduction with the PCA algorithm converts every ten features into one new feature in one sample; there are 4096 samples from the last step, FFT, to proceed with the feature extraction. Then, the new feature is normalized to complete the feature extraction. By the way, normalization is the way to unify the proposed data, which can effectively improve the convergence speed of the algorithm model training.

These two methods engage these algorithms in different sequences to explore optimization. The WPD functionality is mainly for feature extraction, and the PCA functionality can save time on computation.

#### 3.3.2. Build Model

The training model proposed here engaged with four states (normal and abnormal), two directions (horizontal and vertical), two types of sensors, and two modes of signal feature extraction. It aims to develop the model training with high accuracy to distinguish signal features for normal and abnormal bearing states.

The dataset used in this study collects vibration signals from rolling bearings, including normal and abnormal signals, using a self-designed platform, including the four bearing states of normal, misalignment, unbalanced, and impact loads. The original vibration signal is shown in Figure 9 and Figure 10 in the next section, Results and Discussion, and the vibration signal is decomposed with a three-layer wavelet packet. In addition, the Daubechies family (dbN) [30], which exhibits asymmetric, orthogonal, and biorthogonal properties, is used as the candidate mother wavelet function. Additionally, this study uses db11 as the mother wavelet function to obtain the characteristic signal composed of the coefficients of the eight frequency band components in three layers.

The proposed LSTM model utilizes Keras’ deep learning library to conduct the model training with Python. The LSTM algorithm is utilized to classify the normal and abnormal bearing features, and it labels the four types of states with a one-hot mode as the output of the model: [0100] represents normal samples, [0010] represents misalignment samples, [0001] represents unbalanced load samples, and [0000] represents the impact samples. The LSTM model structure is shown in Figure 7 (in Section 3.3).

The Adam optimizer function is chosen to minimize the loss function, and the dropout is set as 0.2 to reduce overfitting. The categorical cross-entropy, a softmax activation plus a cross-entropy loss [31], was used as the loss function for multiclass classification, and the time step was set to 8 as one feature. The laptop with an i7-8550 CPU and a GTX 1050 Ti GPU with 4 GB of RAM performed all the algorithm experiments, and Table 6 lists the LSTM model training parameters. 

## 4. Result and Discussion

The experimental results are gradually present from the vibration signal collection, frequency spectra, feature extraction, and model training of the prediction accuracy. The contents, charts, and segments regarding the sensor model, vibration direction, and types of abnormality depict the sequence arrangement with regard to the apparatus, principles, and methods previously mentioned. 

The data analytics presents vibration signals in the time domain by the type of sensors, ADcmXL3021 and Hi229, in vertical and horizontal vectors across the normal, misalignment, unbalanced, and impact loads. In the frequency spectra, we mainly observe the resolution and dominant frequency in the four types of loads of the two sensors. The two feature extraction methods, 1. PCA-WPD and 2. WPD-PCA, generate eight sub-band segment data in the tables and charts for analytics. The last addresses the training model, the LSTM neural networks argumentation, and the failure mode detection’s accuracy. 

### 4.1. Vibration Signals in Time Domain

With the sensors on the self-design platform, a signal-board computer, Raspberry Pi, collects the vibration data. The time-domain graphics reveal the vibration aspects of the normal, misalignment, unbalanced, and impact loads. These graphs deliver the most basic vibrational information and the background noise filtering capability of the different sensors. Usually, the sensor with better performance can provide higher data collection speed and lower the noise signals. Nevertheless, the cost and performance ratio should be considered issues in most measurement cases, especially in the extensive implementation project.

The following describes the vibration signal patterns in the vertical and horizontal directions, including two different sensors, ADcmXL3021 and Hi229. The graphics cover four-mode vibration signal patterns, including normal, misalignment, unbalanced, and impact loads.

#### 4.1.1. Vertical Direction 

As previously mentioned, the sensors, ADcmXL and Hi229, are attached to the bearing support on the self-design platform, and the vertical direction referred to is shown in Figure 6. In the graphics of ADcmXL3021, Figure 9a, the pattern at the top is the normal state; it presents 120,000 samples in the amplitude range between −2 and 4. The second pattern of the misalignment load shows some unregular peaks, and their amplitude expands to 6. The third one, the unbalanced load, displays a pattern similar to the normal state, but the amplitude range shifts from 0.8 to 1.5. The last pattern generated by the impact load has many significant signal peaks whose amplitudes enlarge to 10 and −10.

Figure 9b contains the four vibration signal patterns in the horizontal direction of Hi229; the sampling number is 120,000, the same as that of ADcmXL3021. By observing these vibration signal patterns of Hi229, we can see that they are pretty similar to each other, except the patterns of the unbalanced load and impact load had some mixed high peaks. 

Compare Figure 9a,b; the ADcmXL3021 provides more precise signal patterns that can help recognize the different types of loads in abnormal states. The misalignment has significant signal peaks with a wider baseband than the impact load. The unbalanced load creates a similar-looking pattern to the normal state, but the amplitude declines about 60%, from 4 to 1.5. The last pattern, impact load, can be most easily detected with its large amplitude and significant signal peaks.

#### 4.1.2. Horizontal Direction

Figure 10 shows the vibration signal graphics in the horizontal direction, including the four patterns of the different loads from the sensors ADcmXL3012 and Hi229. In the same manner, as in the discussion in Figure 9, the graphic of ADcmXL3021 performs better on the raw vibration data collection than Hi229. 

The data from the sensor, ADcmXL3021, on the y-axis and z-axis create the graphics in Figure 9a and Figure 10a, respectively. If we compare the pattern of impact loads, both patterns are similar too. Indeed, the vibration signal pattern could not be precisely the same due to the deviation among the components and the environmental noise in the physical world. 

The primary task in this stage is to confirm that the collected data can classify for load type detection. Here, the raw vibration signals under different loads have been graphed, demonstrating the essential features for judging the type of load. Moreover, the vertical and horizontal patterns are consistent with the trends.

### 4.2. Frequency Spectra of Vibration

Frequency spectra graphics can express the features of a signal at a dominant frequency and harmonics frequency. The frequency-domain graphics can observe the dominant frequency clearly, which can be used as a feature for distinguishing the load types. Converting the signals from the time domain to the frequency domain refers to the principles and equations in Section 2.2. Depicted here are the frequency spectra graphics of the abnormal loads, the misalignment, the unbalanced, and the impact. Comparing the dominant frequencies between the sensors, ADcmXL3021 and Hi229, reveals that the dominant frequencies are coincidentally the same in different loads and are referred to in Figure 11, Figure 12 and Figure 13, as follows.

#### 4.2.1. Misalignment Load 

The frequency spectra graphics of the misalignment loads shown in Figure 11a,b represent the vibration signals in the frequency domain of the sensors, ADcmXL3021 and Hi229, respectively. For the ADcmXL3021 sensor, the dominant frequency of the misalignment load in the vertical and horizontal directions is 149.5 Hz and 144.2 Hz, as in Figure 11a. On the other hand, the graphics of Hi229 show the dominant frequency located at 132.6 Hz and 144.8 Hz. The differences in dominant frequency between ADcmXL3021 and Hi229 in the vertical direction are more significant than in the horizontal direction. 

The amplitude of Hi229 is smaller than that of ADcmX3021, but it would affect the feature extraction process. Even if the Hi229 sensor cannot present very significant features in the time-domain pattern, it can still present an apparent dominant frequency in the frequency-domain graphics. Moreover, the dominant frequency is close enough to the ADcmXL3021 sensor in both the vertical and the horizontal directions.

#### 4.2.2. Unbalanced Load

All the frequency spectra graphics of the unbalanced loads reveal a significant dominant frequency in Figure 12. However, their dominant frequency is less significant in the graphics of the misalignment loads. Figure 12a shows that the dominant frequencies of ADcmXL3021 in the vertical and horizontal directions are 179.3 Hz and 174.5 Hz, respectively. On the other hand, the dominant frequency of the Hi229 sensor is 174.7 Hz in both the vertical and the horizontal directions, as Figure 12b shows.

#### 4.2.3. Impact Load

As well as the previous two loads, the frequency spectra graphics of the impact loads found a significant dominant frequency, as shown in Figure 13a. The ADcmXL3021 graphics display the dominant frequency at 188.5 Hz in both the upper and the lower graphics. The harmonic frequency remains at a much lower amplitude than the dominant frequency. Furthermore, the graphics of sensor Hi229 could not have the outstanding dominant frequency like the ADcmXL3021, but its dominant frequency in the horizontal direction, 187.4 Hz, is pretty close to 188.5 Hz, the ADcmXL3021 dominant frequency.

### 4.3. Feature Extraction Results

The feature extraction processes follow the principles and methods introduced in Section 2 and Section 3 and utilize the dataset generated by the self-design platform. Regarding frequency-domain graphics, the dominant frequency can be an essential feature for judging the types of loads. 

The feature extraction relied on the segments generated by combining three-layer wavelet decomposition (WPD) and principal components analysis (PCA). The methods proposed in Section 3.3 on data processing are method 1, PCA-WPD, and method 2, WPD-PCA, as Figure 8a,b show. The following presents the feature extraction results in the vertical and horizontal directions of the two types of sensors, ADcmXL3021 and Hi229. 

#### 4.3.1. Vertical Direction

Figure 14 and Figure 15 present the charts and segment values of the feature extraction process by methods 1 and 2, which become the datasets for the neural network model training and validation. The charts of the normal state between ADcmXL3021 and Hi229 in method 1 have similar trends in S1, S2, and S3, referring to the blue frame in the upper and lower table in Figure 14. 

Regarding the chart of Hi229, the segments of the abnormal loads and the misalignment, unbalanced, and impact loads, show the unique chart trends in S6, S7, and S8, which provide the various characteristics for the model training, referring to the red frame in the lower table in Figure 14. 

Figure 15 demonstrates the feature extraction adopting method 2 to identify the segments as databases. Similarly, the Hi229 sensor extracted features, S6, S7, and S8, in the red frame of Figure 15, presented significant differences between misalignment, unbalanced, and impact loads. Conversely, the upper chart and table of ADcmXL3021 show almost zero in all segments (S2–S8). The normal state in both charts and segments in the blue frame regarding the ADcmXL3021 and Hi229 sensors remained consistent. 

#### 4.3.2. Horizontal Direction

Figure 16 shows the segments extracted from the frequency-domain data with method 1, PCA-WPD. In the horizontal direction, the data analysis process is the same as the procedure in the vertical direction, but the data sources are different. The chart of ADcmXL3021(upper chart) could not find unique features among the loads, but the chart’s trend is consistent with the HI229 as the numbers in the blue frame. Instead, the HI229 (lower) chart shows the apparent numerical differences in S6 to S8 in the red frame. 

Method 2 extracts feature from the signals data of ADcmXL3021 and Hi229 in the horizontal direction, as shown in Figure 17. The ADcmXL3021 charts (normal, misalignment, unbalanced, impact) display parallelly in S5 to S8, which can be helpful to the training process. The chart of Hi229 shows very different trends in S6–S8 in the red frame to ADcmXL3021. Nevertheless, the normal state in the charts of ADcmXL3021 and Hi229 remained similar for details numbers referring to the blue frame in the tables. 

### 4.4. Model Training Results and Accuracy

The training model performs the testing and validation by the algorithm, data process, and model training parameter (Table 6), as mentioned in previous Section 2.4 and Section 3.3). Table 7 presents the accuracy of the testing results with method 1 and method 2, with two sensors in the vertical and horizontal directions. The best accuracy result is 97.35%, achieved with method 2, with the Hi229 sensor in the vertical direction. 

#### Loss and Accuracy

The experiment of model training only performs the LSTM neural network with the dataset we created. To prove the model is appropriate for the applications, it might require conducting different neural networks to verify the performance, such as in the work conducted by Wei You et al., 2020 [32], or checking the loss function and accuracy. Here we adopt the cross-entropy loss function for validation.

Figure 18a plots the training loss curves of the LSTM neural network about method 2, Hi229, in the vertical direction. The validation curve does not show stability as well as the training curve, but its scatter points still fit the main trends of the training curve. The training curve (blue line) converged quickly in ten epochs and showed minimal ripples, except for the singularity near the 60 epochs.

The plot of the LSTM model (method 2, Hi 229, vertical direction) in accuracy curves is opposite to the loss curves. The training curve pitched up to approach 98% (or 0.98 in the graphic) accuracy, and it converged quickly in ten epochs and became a smooth straight line, as Figure 18b shows. The validation curve has more scatter points but remains within 96% accuracy. 

## 5. Conclusions

This work achieved failure mode detection with up to 97% probability of the feature extraction method (WPD-PCA), the LSTM neural network, and a common sensor, Hi229. It contributes to the more significant amount of sensor deployment scenarios and provides a paradigm of bearing vibration analysis for relevant research. 

In general, the high-end sensors can perform better than the low-cost sensors, but through feature extraction and model training, the low-cost sensors can also deliver critical messages. For instance, the accelerometer sensor AdcmXL3021 has a better signal resolution in the time domain than sensor Hi229 in all states (normal and abnormal states). Nevertheless, when the vibration data convert into the frequency spectra graphics, the dominant frequency of both sensors is quite similar in our experiments. It narrows the gap between high-performance and typical low-cost sensors in applications such as bearing vibration measurement. While typical low-cost sensors have lower sensitivity and noise-filtering capabilities, there appear to be fewer effects in the subsequent feature extraction process. The evidence can be seen in the best results in the accuracy of method 2, Hi 229, in the vertical direction.

The external impact load does not significantly affect the bearing response compared with the misalignment and unbalanced loads generated by the machines. Regarding the frequency spectra, the dominant frequency of the impact load is around 188.5 Hz to 170.8 Hz, as Figure 13 shows. On the other hand, the dominant frequency of the misalignment and unbalanced loads, at the lowest and the highest, is 132.6 Hz and 179.3 Hz, as shown in Figure 11b and Figure 12a. The impact load dominant frequency is a little higher than the other two abnormal states but still in the range of the bearing anomaly feature frequency of BPFI and BPFO, as previously mentioned in Section 2.1. 

Proper feature extraction and neural network algorithm can improve computing efficiency and accuracy in failure mode prediction. The actual machine operation state could combine the unexpected external loads and the load generated by misalignment and unbalanced loads from the machines. There are more sophisticated situations we cannot wholly envisage and present in this study. However, we create a bearing failure mode detection paradigm that can serve as a reference for bearing life studies in more advanced areas.

In future works, we can mix three failure modes to emulate the states of actual mechanical operations. For example, there could be a combination of misalignment and unbalanced load, including periodic impact load. Nowadays, many studies intend to acquire the signals from the actual machine for analysis, such as a gearbox that could be a more straightforward approach to the problems. Conversely, our approach is bottom-up, stacking up the signals with various loads that the self-design platform can generate, using the combination of these signals to analyze the influence of the failure mode on bearing life. 

## Figures and Tables

**Figure 1 sensors-22-06167-f001:**
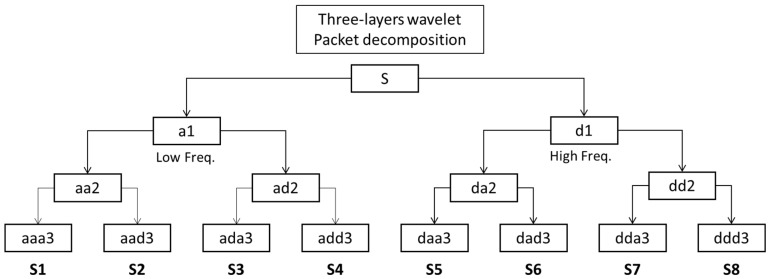
Wavelet packet decomposition diagram (three layers).

**Figure 2 sensors-22-06167-f002:**
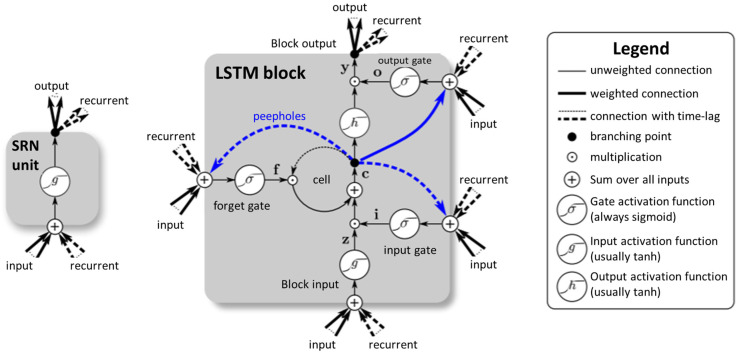
Simple recurrent network (SRN) and LSTM schematic diagram (Klaus Greff et al.) [24].

**Figure 3 sensors-22-06167-f003:**
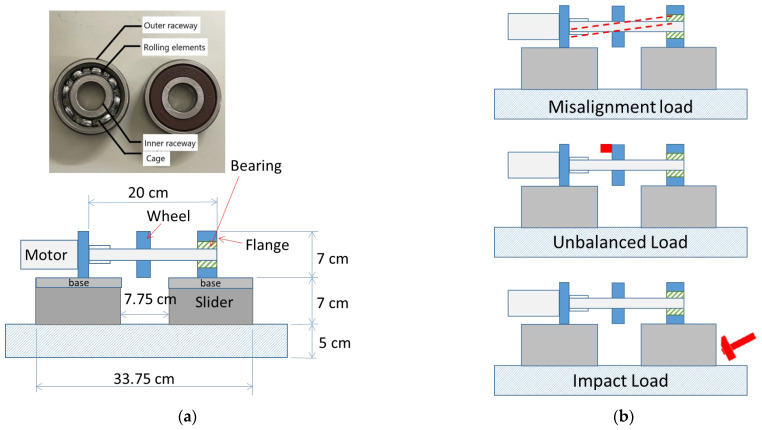
Self-design platform: (**a**) dimension and bearing pictures; (**b**) illustration of loads, including misalignment load, unbalanced load, and impact load.

**Figure 4 sensors-22-06167-f004:**
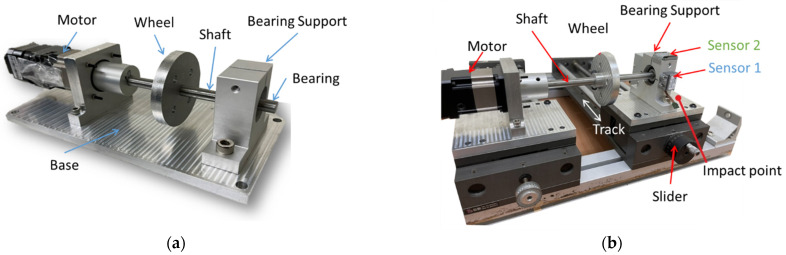
Platforms for collecting vibration signals data: (**a**) Platform 1: for normal state; (**b**) Platform 2: for abnormal state.

**Figure 5 sensors-22-06167-f005:**
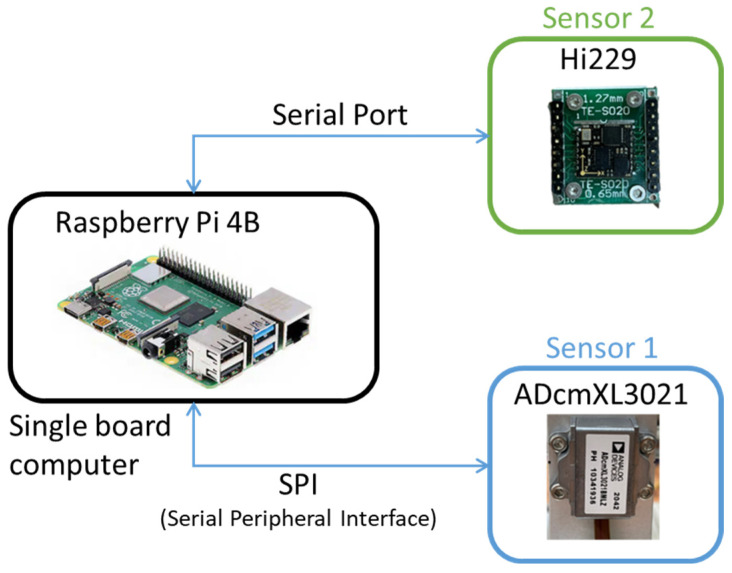
Apparatus for vibration signals collection.

**Figure 6 sensors-22-06167-f006:**
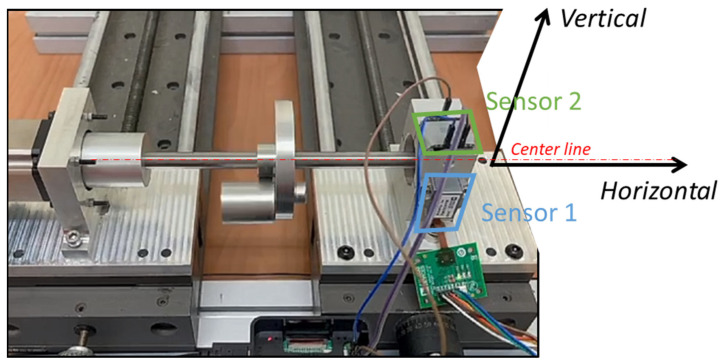
Sensor settlement on the self-design platform (with unbalanced load).

**Figure 7 sensors-22-06167-f007:**
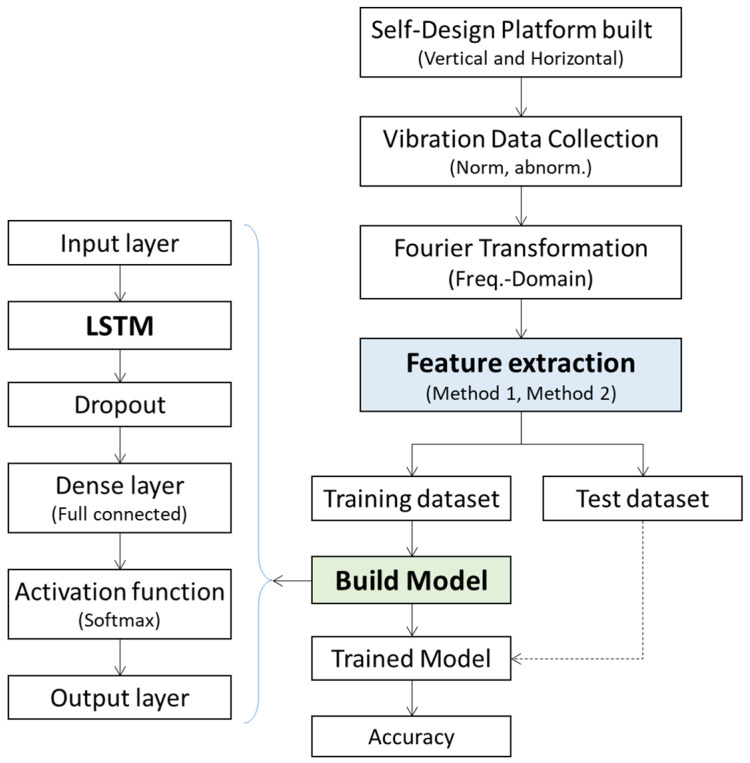
Data analysis flow chart.

**Figure 8 sensors-22-06167-f008:**
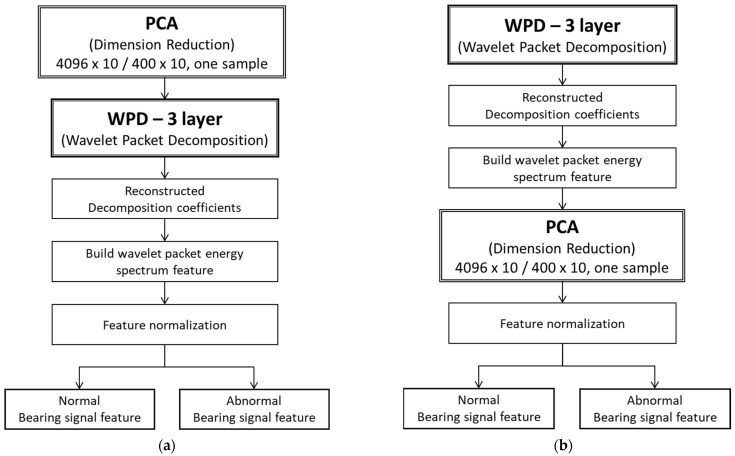
Feature extraction methods: (**a**) PCA-WPD method; (**b**) WPD-PCA method.

**Figure 9 sensors-22-06167-f009:**
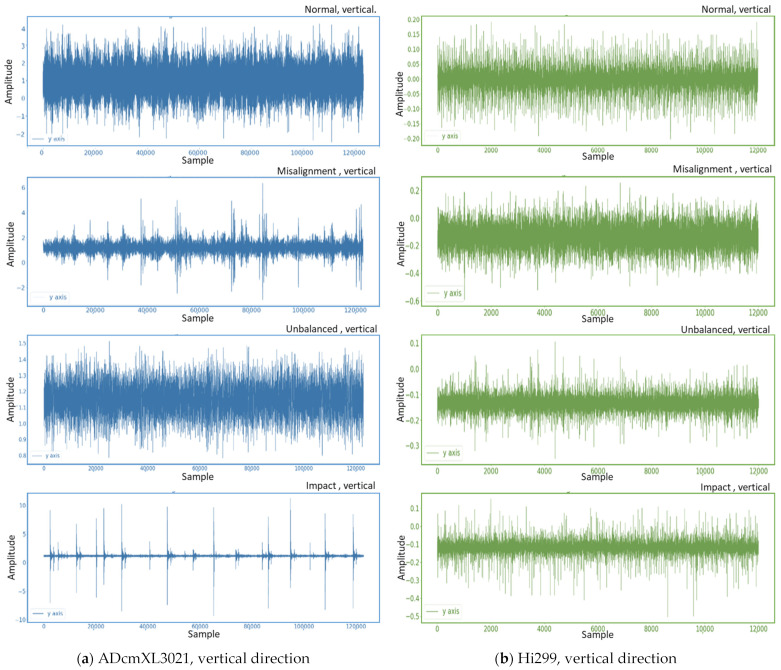
Vibration signal graphics in the vertical direction: (**a**) ADcmXL3021, signal patterns of normal, misalignment, unbalanced, and impact loads; (**b**) Hi229, signal patterns of normal, misalignment, unbalanced, and impact loads.

**Figure 10 sensors-22-06167-f010:**
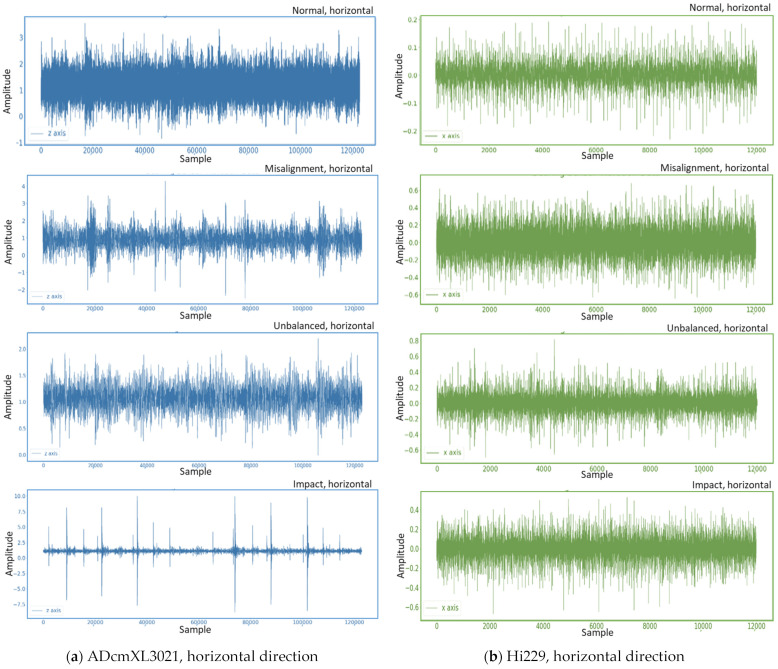
Vibration signal graphics in the horizontal direction: (**a**) ADcmXL3021, graphics of normal, misalignment, unbalanced, and impact loads; (**b**) Hi229, graphics of normal, misalignment, unbalanced, and impact loads.

**Figure 11 sensors-22-06167-f011:**
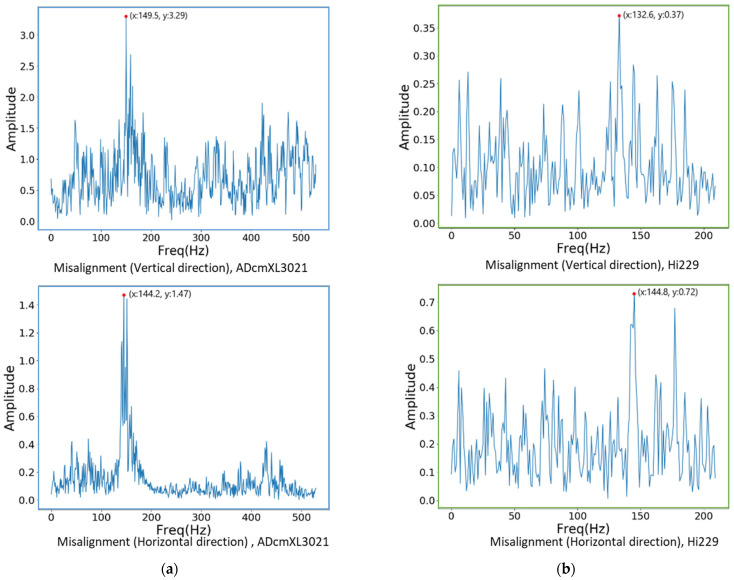
Frequency spectra graphics of the misalignment loads: (**a**) ADcmXL3021, the vertical direction (upper) and the horizontal direction (lower); (**b**) Hi229, the vertical direction (upper) and the horizontal direction (lower).

**Figure 12 sensors-22-06167-f012:**
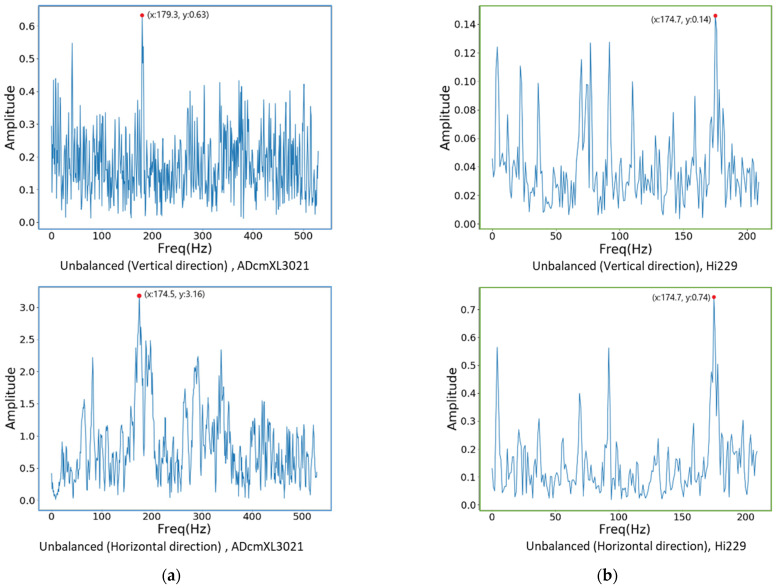
Frequency spectra graphics of the unbalanced loads: (**a**) ADcmXL3021, the vertical direction (upper) and the horizontal direction (lower); (**b**) Hi229, the vertical direction (upper) and the horizontal direction (lower).

**Figure 13 sensors-22-06167-f013:**
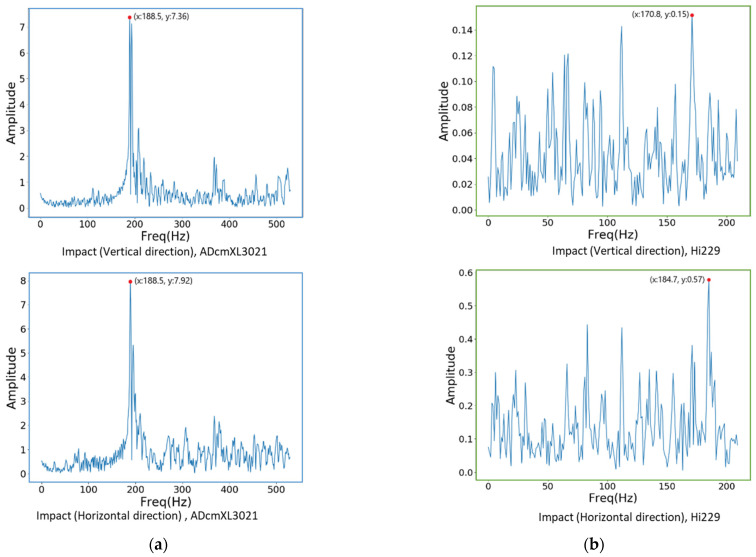
The vibration signal in the frequency domain under impact loads: (**a**) ADcmXL3021, the vertical direction (upper) and the horizontal direction (lower); (**b**) Hi229, the vertical direction (upper) and the horizontal direction (lower).

**Figure 14 sensors-22-06167-f014:**
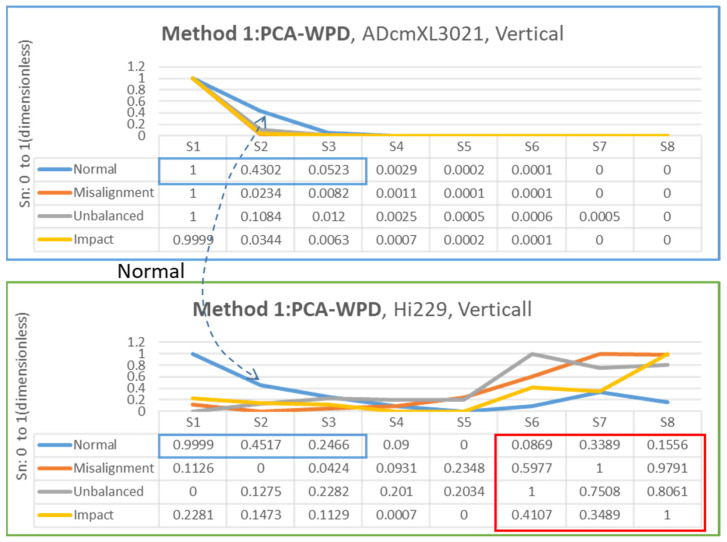
Chart and segment value of feature extraction for method 1: PCA-WPD, ADcmXL3021 upper, Hi229 lower; vertical direction.

**Figure 15 sensors-22-06167-f015:**
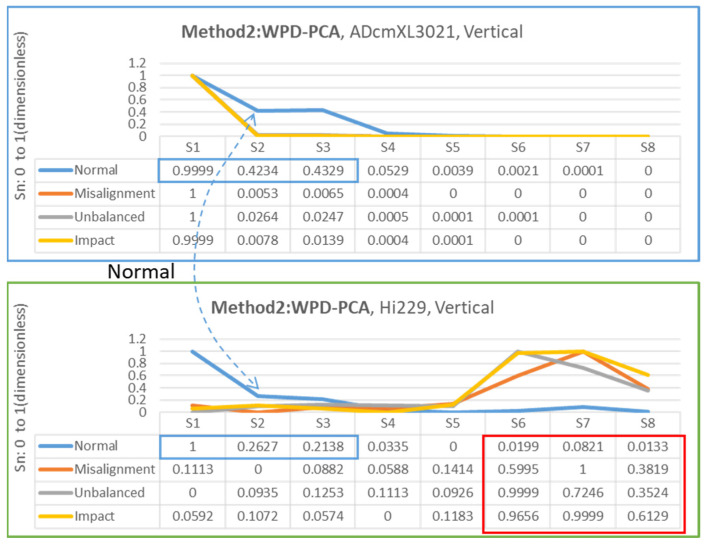
Chart and segment value of feature extraction for method 2: WPD-PCA, ADcmXL3021 upper, Hi229 lower; vertical direction.

**Figure 16 sensors-22-06167-f016:**
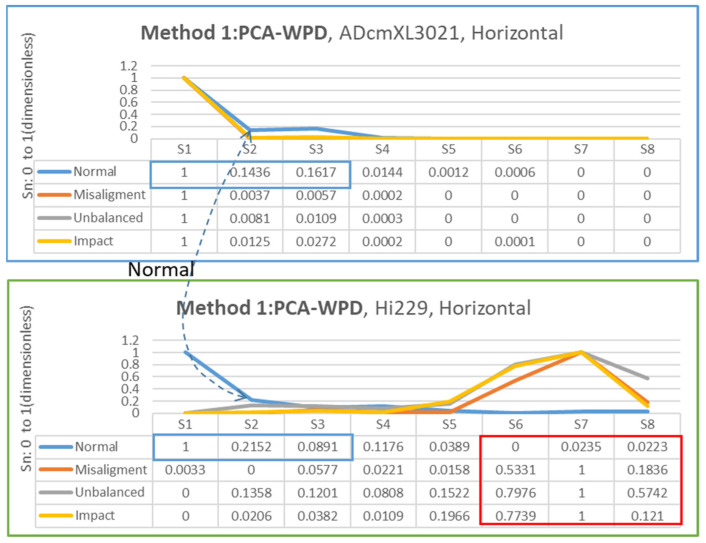
Chart and segment value of feature extraction for method 1: PCA-WPD, ADcmXL3021(upper), Hi229(lower); horizontal direction.

**Figure 17 sensors-22-06167-f017:**
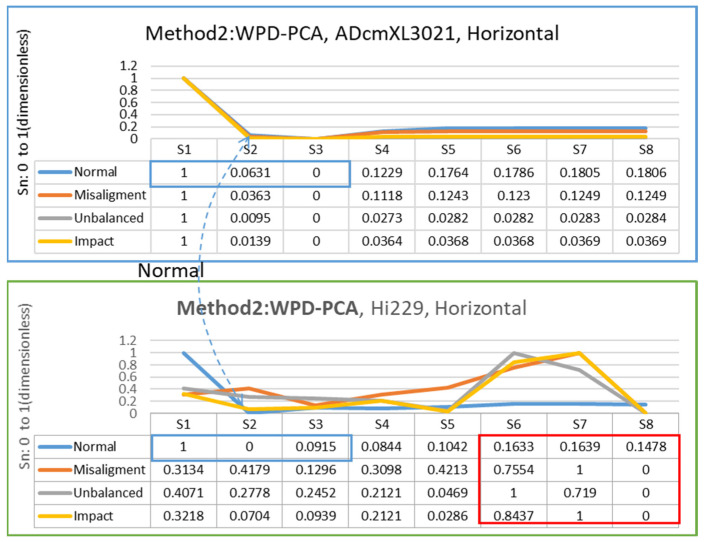
Chart and segment value of feature extraction for method 2: WPD-PCA, ADcmXL3021(upper), Hi229(lower); horizontal direction.

**Figure 18 sensors-22-06167-f018:**
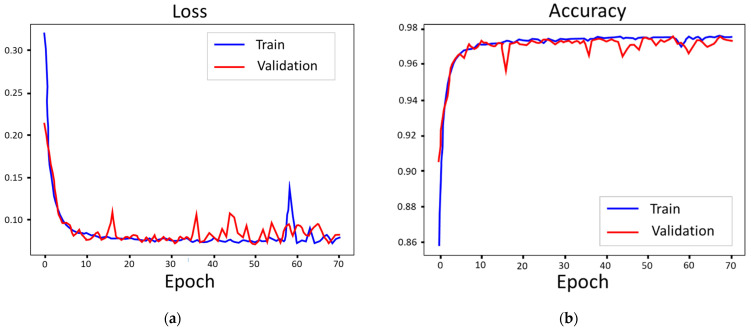
Loss and accuracy curves of the LSTM model (method 2, Hi229, vertical direction): (**a**) loss; (**b**) accuracy.

**Table 1 sensors-22-06167-t001:** Samples of vibration signal datasets.

Dataset	Accelerator and DAQ	Sensor Type	Number of Sensors	Fault Type	Fault Mode	Characteristic	Reference
Paderborn University Dataset	Model No. 336C04, PCB Piezotronics, Inc.	Accelerometer Current sensor Thermocouple	1 2 1	1. inner race wear 2. outer race wear 3. lifetime	Artificial damages/accelerated lifetime tests	Simple vibration signals and MCS using sensor fusion	[25]
CWRU Dataset	N/A	Accelerometer	2	1. inner race wear 2. outer race wear 3. ball wear	Artificial damages	Multiple bearings of different sizes	[26]
IMS Dataset	National Instruments DAQ Card™-6062E data acquisition card	Accelerometer	2	1. inner race wear 2. outer race wear 3. lifetime	Natural	Natural bearing defect evolution runs over long durations	[27]
Pronostia Dataset	N/A	Accelerometer Thermocouple	2 1	lifetime	Natural	Actual data about the accelerated bearing degradation at varying operating conditions	[28]
XJTU-SY Bearing Dataset	PCB 352C33	Accelerometer	2	lifetime	Artificial damages	Three different radial forces are used to accelerate bearing service life	[29]
* Experimental Dataset	1. ADcmXL3021 2. Hi229	Accelerometer Accelerometer	1 1	1. misalignment 2. unbalanced load 3. impact	Artificial damages	Two different accelerometers and experiments with three fault modes	Our work

* Two datasets are generated from the accelerometer ADcmXL3021 and Hi229, respectively.

**Table 2 sensors-22-06167-t002:** Dataset amount.

States	ADcmXL3021 Sensor	Hi229 Sensor
normal	65,674 points	50,492 points
misalignment	21,587 points	25,276 points
unbalanced load	21,831 points	32,903 points
impact	19,407 points	18,330 points

**Table 3 sensors-22-06167-t003:** Nominal bearing dimensions.

Structure	Parameter
Inner raceway diameter	10 mm
Outer raceway diameter	30 mm
Width	9 mm
Rolling-element diameter	5.6 mm
Pitch diameter	20 mm
Contact angle	0°
Number of rolling elements ^1^	8

^1^ Bearing type: deep groove ball bearing.

**Table 4 sensors-22-06167-t004:** Loading in experiments.

Condition	Load	Load/Angle
Normal	Flange	101.3 g/0°
Misalignment	Flange	101.3 g/1.29°
Unbalanced	Flange + Load	101.3 + 51.54 g/0°
Impact	Flange	101.3 g/0°

**Table 5 sensors-22-06167-t005:** Specifications of AdcmXL3021 and Hi229.

Performance Metric	ADcmXL3021 ^1^	Hi229 ^2^
Measurement range	±50 g (unit: mg/LSB)	±8 g (unit: G)
Sample rate	3.439 kHz	400 Hz
Maximum linear acceleration	N/A	0 to 115$\;m/s^{2}$
Cross Axis Sensitivity	2%	N/A
Nonlinearity	±0.2 to ±1.25	±0.5%
Sensor Resonant Frequency	21 kHz	N/A
Temperature range	−40 °C to +105 °C	−20 °C to 85 °C
Cost (USD)	USD 269.53	USD 35

^1^ AdcmXL3021 datasheet: https://www.analog.com/media/en/technical-documentation/data-sheets/adcmxl3021.pdf, accessed on 16 Aug 2022.; ^2^ Hi229 datasheets: https://www.hipnuc.com/en/product_hi229.html, accessed on 16 Aug 2022.

**Table 6 sensors-22-06167-t006:** Model training parameters.

Structure	LSTM
LSTM neuron 1	256
Epochs	70
Optimizer	Adam
LR	0.001
Dense	4
Dropout	0.2
Activation	softmax

**Table 7 sensors-22-06167-t007:** Accuracy of testing results.

Proposed Method	Accuracy (%)
Method 1-ADcmXL3021-vertical direction	94.88
Method 1-ADcmXL3021-horizontal direction	89.31
Method 1-HI229-vertical direction	81.50
Method 1-HI229-horizontal direction	90.86
Method 2-ADcmXL3021-vertical direction	93.30
Method 2-ADcmXL3021-horizontal direction	87.57
**Method 2** **-** **HI229-vertical direction**	**97.35**
Method 2-HI229-horizontal direction	96.51

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
