# Peer review of "Failure Mode Detection and Validation of a Shaft-Bearing System with Common Sensors"

_sensors, 2022, doi:10.3390/s22166167_

Round 1

Reviewer 1 Report

Please find my comments in the attached document. 

Author Response

First of all, thanks for your constructive comments. Please find the attached file for tracking changes, and we put the annotation in the line number regarding all reviewer's questions, including yours, that can help to explain why we do the changes. 
The reply to your question is as follows. We could misunderstand some questions and reply with meaningless answers, please be understanding.

  1. Line 46: Define RUL in the first instant.

Reply:

Thanks for the suggestion and revised in line 52. ( the original line 46)

  1. Line 48: Each of the paragraphs should not only discuss the literature but also point out how the present research is going to address the gap areas.

Reply:

Thanks for the notification. We did not notify this shortage the first time.

In the literature review, we add extension information in line 50, 55, 64, 80, 127, to explain more about the gaps in technology. 

  1. The flow among the paragraph ending at line 62 and the one starting from line 63 is not clear. Please figure out and explain the connection between them.

Reply:

Thanks for raising this concern. We removed these paragraphs from lines 70 to 75, including the [7] reference. 

  1. The end of the introduction should point to the objective of the paper, and its motivation. As of now, the structure of the paper makes it difficult to decipher the intent of the authors.

Reply:

Yes, this is important. We add two paragraphs to explain our target, in lines 130 to 141.

  1. Please provide the structure of the paper before moving to the next section.

Reply:

Thanks for this reminder. We add the sections briefing in lines 142 to 148.

  1. The novelty of the work is not evident in the introduction section. this should be clearly mentioned, along with the motivation for carrying out this work.

Reply:

Thanks for this reminder. We add the sections briefing in lines 136 to 141.

  1. In recent times, eigen perturbation based single-sensor condition monitoring for bearings has been carried out in real-time. As the present approach does not consider a real-time structure, please provide reasons as to why this method should be adopted in the literature

Reply:

Thank you for delivering a new concept to us.

All AI neural network solutions do require time for data collection and stimulation. Therefore, it may be impractical to faults detection on large dynamic structures or machines due to excessive measurements and computations. The eigenvalue perturbation-based method for the bearing can save time on data collection and stimulation in AI solutions. Therefore, there should be more and more bearing fault detection using eigenvalue perturbation in the future.

(This part is not in the manuscript.)

Here are the references

  1. Yu, L. Cheng, L.H. Yam, Y.J. Yan, Application of eigenvalue perturbation theory for detecting small structural damage using dynamic responses, Composite Structures, Volume 78, Issue 3, 2007, Pages 402-409, ISSN 0263-8223, https://doi.org/10.1016/j.compstruct.2005.11.007.

  1. Section 2.2: FFT is a very common textbook topic which I believe should either be omitted completely, or made concise for brevity, or put in an Appendix altogether.

Reply:

Yes, we agree with this point. We deleted all equations but kept the description and FFS reference, in line 192.  

  1. Out of the gamut of feature extraction process and approaches, why has a wavelet-based method been adopted in this paper? What are the shortcomings of other methods – say in particular – recursive residual errors, eigen vector change, or even time-varying auto-regressive models? The authors must justify their claims with sufficient evidence.

Reply:

This is a really good question but not easy to answer, too. In mathematics, we know a wavelet series represents a square-integrable function by a specific orthonormal series generated by a wavelet. We learn that the Wavelet Packet Decomposition (WPD) is a wavelet transform where the discrete-time (sampled) signal is passed through more filters than the discrete wavelet transform. WPD has been widely used in speech and music discrimination.

(This part is not in the manuscript.)

  1. On a similar note, PCA is a textbook topic which I believe should be made concise or moved to an Appendix.

Reply:

Thanks for raising this concern. We made it more concise.  

  1. While considering PCA, it should be understood that this is an offline process and therefore, the proposed approach will never function online. Real-time recursive PCA approaches have been abundantly used that employ eigen perturbation strategies for bearing monitoring and defect detection. The authors should justify the use of PCA here over RPCA.

Reply:

Thanks for this concern. We’ve cited the Recursive Principal Components Analysis article as a reference in line 291.

  1. Please increase the size of the fonts in Figure 2.

Reply:

Yes, increased word size up to 16 pt.

  1. Compared to existing approaches that function in real-time – such as eigen perturbation, Kalman filter, and diagonalization – the present method does not function online and provides a more computationally demanding structure. Please provide a computational expense overview of the proposed approach, and compare it against existing methods.

Reply:

Thanks, I appreciate this question but need more time to study and think them over. Would you please provide some references for us? So we can learn faster, thank you.

(This part is not in the manuscript.)

  1. Before moving to section 4.3, please explain how the method will remove the noise components that are evident from each of the Fourier spectra. In recent times, single-sensor based real-time condition monitoring of bearings has been carried out by eigen perturbation based SSA. The authors must identify the potential advantage of using the proposed approach over real-time SSA and justify it with sufficient evidence.

Reply:

Thanks for this question. It’s good, but not easy for us to answer immediately. We need more time to study the eigen perturbation-based SS and if SSA stands for single-scattering albedo. Can you provide some references for us to study?

(This part is not in the manuscript.)

Reviewer 2 Report

I found your article very interesting, but in my opinion below remarks would improve your manuscript under the scientific level.

Comments and Suggestions for Authors:

1.       At first I would like to refer to the terminology, I’m not sure if you should use term “shaft bearing”. In my opinion I would change it to “shaft-bearing system” or “rotor based system”. In the current form it sounds very plain without any precise context.

2.        The same with “off-center”, I would call it misalignment. The same in Figure 3, this is the classical shaft’s misalignment situation

3.       In the beginning of Introduction I miss the information on other features having impact on the bearing’s life such as friction torque or radial internal clearance. I suggest to add 2 following references discussing this problem:

·       Tong V.-C et al. (2018), Study on the running torque of angular contact ball bearings subjected to angular misalignment, Proceedings of the Institution of Mechanical Engineers, Part J: Journal of Engineering Tribology, 232, 890-909.

·       Ambrożkiewicz et al. (2022), The influence of the radial internal clearance on the dynamic response of self-aligning ball bearings, Mechanical Systems and Signal Processing, 171, 108954.

4.       What is “alcoholic EGG” in line 65?

5.       Subsection 2.1, mentioned frequencies are not anomaly frequencies, they correspond to characteristic frequencies, and they can’t be called anomaly one. The amplitude of peaks determines if there is damage on specific element or not. In any case, characteristic frequencies are observable in the frequency spectra.

6.       For what bearing, the characteristic frequencies are calculated?

7.       Sentence 153 is incorrect, the noise will never bury characteristic peaks of bearing if the measurement system is calibrated and well assembled. Please remove this sentence or I expect its explanation, what the Author meant?

8.       I suggest to give only information on FFT, which refers to calculation of characteristic frequencies. I don’t see the reason for explaining IFR.

9.       Referring to feature extraction I suggest to cite the following paper, which is a rich source of knowledge about the diagnostics features used in the diagnostics of rotating machines: Sharma V. et al. (2016), A review of gear fault diagnosis using various condition indicators. Procedia Engineering 144, 253-263.

10.   I appreciate the collection of public datasets in subsection 2.5, I would suggest to emphasize it.

11.   Table 3, please correct Thickness -> Width, Rolling-element diameter, Bearing pitch diameter -> Pitch diameter, Rolling element amounts -> Number of rolling-elements, Contact angle -> Pressure angle (both are correct). What kind of bearing it is, what is its type?

12.   What are your features for PCA and ML model? This is very important information. How they are obtained from time-series?

13.   How the model is obtained, what is the number of samples for training and validation. I didn’t find it in the manuscript.

14.   I would suggest to correct the first introducing sentence to Conclusions. It is not important to name the sensor, however to emphasize its better diagnostics capabilities.

15.   In Conclusions I miss the sentence about your future works.

Author Response

First of all, thanks for your constructive comments. Please find the attached file for tracking changes, and we put the annotation in the line number regarding all reviewer's questions, including yours, that can help to explain why we do the changes. 
The reply to your question is as follows. We could misunderstand some questions and reply with meaningless answers, please be understanding.

Please use “Tracking change.” to read the attached file. 

  1. At first I would like to refer to the terminology, I'm not sure if you should use term "shaft bearing". In my opinion I would change it to "shaft-bearing system" or "rotor based system". In the current form it sounds very plain without any precise context.

Reply:

Thanks for this suggestion. We change the title to "Failure Mode Detection and Validation of a Shaft-Bearing System with Common Sensors."

  1. The same with "off-center", I would call it misalignment. The same in Figure 3, this is the classical shaft's misalignment situation

Reply:

Thanks for the suggestion. We changed "off-center" to "misalignment" in all contents and figures (starting at Line 32, a total of 36 places changed), including Figure 3.

  1. In the beginning of Introduction I miss the information on other features having impact on the bearing's life such as friction torque or radial internal clearance. I suggest to add 2 following references discussing this problem:
  • [32]Tong V.-C et al. (2018), Study on the running torque of angular contact ball bearings subjected to angular misalignment, Proceedings of the Institution of Mechanical Engineers, Part J: Journal of Engineering Tribology, 232, 890-909.
  • [33]Ambrożkiewicz et al. (2022), The influence of the radial internal clearance on the dynamic response of self-aligning ball bearings, Mechanical Systems and Signal Processing, 171, 108954. 

Reply:

Yes, the points you mentioned are essential to most readers, and thanks for sharing the references. In the Introduction section, you can read these two references as complements to the misalignment issue in Line 34 – 39.  

  1. What is "alcoholic EGG" in line 65?

Reply:

Thanks for highlighting this mistake, and sorry to make you confused.

The correct nominal is "alcoholic electroencephalograph (EEG)," please check line 65 for the correction.    

  1. Subsection 2.1, mentioned frequencies are not anomaly frequencies, they correspond to characteristic frequencies, and they can't be called anomaly one. The amplitude of peaks determines if there is damage on specific element or not. In any case, characteristic frequencies are observable in the frequency spectra.

Reply:

Thanks for raising the concern; it helps solidify the foundation of this work.

We changed "Subsection 2.1, bearing anomaly feature frequency" to "bearing characteristic frequency. I also added the statement based on your comments," the bearing characteristic frequencies and the amplitude of peaks determines if there is damage to a specific element or not. In any case, characteristic frequencies are observable in the frequency spectra.", in line 159 to 161.  

  1. For what bearing, the characteristic frequencies are calculated?

Reply:

Ball bearing, check details in the article by Akhand Rai (2015).

In our work, the bearing for the test is a conventional deep groove ball bearing, as Figure 3(a) shows, in line 358, and the nominal bearing dimension is listed in Table 3. 

  1. Sentence 153 is incorrect, the noise will never bury characteristic peaks of bearing if the measurement system is calibrated and well assembled. Please remove this sentence or I expect its explanation, what the Author meant?

Reply:

Thanks for notifying this point. We removed this sentence in line 153, and I appreciate this reminder. Please check the contents revised in line 180.  

  1. I suggest to give only information on FFT, which refers to calculation of characteristic frequencies. I don't see the reason for explaining IFR.

Reply:

Thanks for raising the concern; we removed most descriptions of FFT, the part of IFR, and equations, revised in line 185. 

  1. Referring to feature extraction I suggest to cite the following paper, which is a rich source of knowledge about the diagnostics features used in the diagnostics of rotating machines: Sharma V. et al. (2016), A review of gear fault diagnosis using various condition indicators. Procedia Engineering 144, 253-263.

Reply:

Reply: Yes, it is like your recommendation, and we cited it in lines 201 to 204.

  1. I appreciate the collection of public datasets in subsection 2.5, I would suggest to emphasize it.

Reply:

Thanks for your appreciation. We write a paragraph to emphasize the usage, in line 333 to 336.  

11.Table 3, please correct Thickness -> Width, Rolling-element diameter, Bearing pitch diameter -> Pitch diameter, Rolling element amounts -> Number of rolling-elements, Contact angle -> Pressure angle (both are correct). What kind of bearing it is, what is its type?

Reply:

Thanks for the suggestion. We correct all items as recommended. And bearing type is deep groove ball bearing.  

12.What are your features for PCA and ML model? This is very important information. How they are obtained from time-series?

Reply:

Thank you for raising this concern. 

The newly added contents in lines 460 to 469 provided signal composed features and defined output model [0100],[0010],[0001]……。

13.How the model is obtained, what is the number of samples for training and validation. I didn't find it in the manuscript.

Reply:

Thanks for raising the concern.

Please refer to the flow chart in line 416, the contents mentioned above in Q12, and the dataset in line 342.

The process starts with approximately 12,000 samples of each mode, and the Dataset amount shows in Table 2. Then, the FFT converts to frequency spectra and processes the feature extraction regarding the definition of the feature as aforementioned in Q12. There are 4,096 samples processed in this work. Please check lines 439 to 459. The contents have been revised to be precise.

14.I would suggest to correct the first introducing sentence to Conclusions. It is not important to name the sensor, however to emphasize its better diagnostics capabilities.

Reply:

Thanks for raising the concern. We rewrite the sentence as per your suggestion in lines 676 to 679.

  1. In Conclusion I miss the sentence about your future works.

Reply:

Thank you for the reminder. We add the future work paragraph in lines 706 to 712.

Round 2

Reviewer 1 Report

The authors have partially addressed my review comments. In this round, the following responses to the previous queries are requested: 

1. Previous query #7: When a query is presented, it is usually a norm to include the content in the revised version of the manuscript. The response to this query was informal. Rather, it is to my observation that the authors have not adequately addressed this concern. While the proposed approach does not cater to real-time formulation, the authors should justify its inclusion considering the dearth of online approaches - especially using eigen perturbation. 

2. Previous query #7: The authors also mention that 'there should be more and more bearing fault detection using eigenvalue perturbation in the future.' If this is the case, the paper should present applications or mention noticeable instances of this approach. The authors must justify their thoughts here. 

3. Previous query #9: This question has not been adequately responded to. A comparative study of the methods asked, or at least a qualitative discussion as to why wavelet-based methods should be used instead of recursive residual errors and eigenvector changes must be provided in the revised manuscript. 

4. Previous query #13: The response to this question is highly informal. IT is the prerogative of the authors to go through a detailed literature review that includes the study of eigen perturbation approaches. It is asked to provide an overview - and not a comparison - of the proposed approach and the existing eigen perturbation and Kalman filter strategies. This should not include any new experimentation but involve a detailed understanding of the scientific content. 

5. Previous query #13: The reviewer apologizes for not including the abbreviation expansion in the first instance. Singular spectrum analysis (SSA) has been recently carried out in real-time for condition monitoring of bearings. Please explain how the proposed method will remove the noise components that are evident from each of the Fourier spectra which are explicitly done by SSA in a more efficient manner - without the use of any additional external filters. 

Author Response

This revision removes the redundant information from the introduction section and also slightly rearranged the citation, as the general bearing lifetime studies in [1]-[6], feature extraction, and neural network in [7]-[16] to make the article more comprehensible.

Also, we removed the citation of Eigen perturbation from the manuscript because in-time bearing monitoring could not be the preliminary issue in the study of failure mode detection for bearing lifetime prediction. Generally, the bearing life can be as long as several years, and even the worn-out stage can last for many months, as the citation [6] depicted in line 50.

We appreciate your introducing the Eigen perturbation and SSA methods that were not in our original manuscript. That helps us to learn more about state-of-art technology in this field. Also, thanks for the constructive comments on the other part of our manuscript. The point-by-point response to your query is as follows.

Best regards,

Shuhao

Reviewer 2 Report

All remarks have been introduced to improve the manuscript. I recommend its publishing in the present form.

Author Response

This revision removes the redundant information from the introduction section and also slightly rearranged the citation, as the general bearing lifetime studies in [1]-[6], feature extraction, and neural network in [7]-[16] to make the article more comprehensible. We also improved some statements in the abstract. Finally, thanks for those instructive comments that help a lot with the manuscript quality improvement.

Best regards,

Shuhao
